# Application of Near-Infrared Spectroscopy to Investigate Some Endogenic Properties of *Pleurotus ostreatus* Cultivars

**DOI:** 10.3390/s20226632

**Published:** 2020-11-19

**Authors:** Marietta Fodor, Erika Etelka Mikola, András Geösel, Éva Stefanovits-Bányai, Zsuzsanna Mednyánszky

**Affiliations:** 1Institute of Food Quality, Safety and Nutrition, Department of Applied Chemistry, Faculty of Food Science, Szent István University, Villányi str 29-43, H-1118 Budapest, Hungary; Mikola.Erika.Etelka@szie.hu (E.E.M.); Stefanovitsne.Banyai.Eva@szie.hu (É.S.-B.); 2Institute of Sustainable Horticulture, Department of Vegetable and Mushroom Growing, Faculty of Horticulture Science, Szent István University, Villányi str 29-43, H-1118 Budapest, Hungary; Geosel.Andras@szie.hu; 3Institute of Food Quality, Safety and Nutrition, Department of Food Chemistry and Nutrition, Faculty of Food Science, Szent István University, Somlói út 14-16, H-1118 Budapest, Hungary; Mednyanszky.Zsuzsanna@szie.hu

**Keywords:** *Pleurotus ostreatus*, amino acid content, TPC, FRAP, FT-NIR, spectra profiling, qualification, chemometrics, PCA, LDA

## Abstract

Fourteen different *Pleurotus ostreatus* cultivars (Po_1–Po_14) were tested for free amino acid content (fAA), total polyphenol content (TPC), and antioxidant capacity (Ferric Reducing Ability of Plasma—FRAP) to select the cultivars with the most favorable traits. Automatic amino acid analyzer (fAA) and spectrophotometric assay (TPC, FRAP) results as well as Fourier-transform near infrared (FT-NIR) spectra were evaluated with different chemometric methods (Kruskal–Wallis test, Principal Component Analysis—PCA, Linear Discriminant Analysis—LDA). Based on total free amino acid concentrations and FRAP values, the Po_2 cultivar was found to be the most favorable. Types Po_3, Po_8, Po_10 and Po_12 were separated using PCA. Based on the spectral profile, they may contain polyphenols and reducing compounds of different qualities. LDA classification that was based on the concentrations of all free amino acids, cysteine, and proline of the cultivars was performed with an accuracy of over 90%. LDA classification that was based on the TPC and FRAP values was performed with an accuracy of over 83%.

## 1. Introduction

New challenges in food science aim to provide a scientific basis for modern food production, in particular with regard to current nutritional needs, consumer health protection and environmentally friendly techniques.

The scientific investigations are based on state-of-the-art analytical research, bearing in mind the beneficial physiological effects of endogenous components with reducing properties that exceed the body’s energy and nutrient requirements [1].

Due to the increasing consumer demand for bioactive mushrooms, it is necessary to grow resistant varieties in which the macronutrients and antioxidant components essential for vital functions are present in sufficient amounts [2,3,4].

The most widely consumed mushroom in the world is *Agaricus bisporus*, followed by *Pleurotus* spp. and *Lentinula edodes*. They are easy to grow and have a high nutritional value [1]. *Pleurotus* mushrooms are healthy foods, low in calories and in fat, rich in protein, carbohydrates, chitin, minerals (calcium, phosphorus, iron) and vitamins (thiamin, riboflavin and niacin) They also contain high amounts of γ-aminobutyric acid (GABA) and ornithine [5].

The bioactive compounds of mushrooms are responsible for antioxidant, radical scavenging, antiviral and antibacterial properties. Examples of such bioactive compounds are polysaccharides, dietary fiber, and antioxidants (vitamins C, B12 and D; folate; tocopherols, ergothioneine, carotenoids and polyphenols). Synergistic effect of these molecules may stimulate the free radical scavenging activity of these mushrooms [6]. Reducing components (phenolics and polyphenolics) [7,8] of mushrooms are formed during secondary metabolic processes that inhibit or delay the effects of oxidative damage and are involved in protection against free radicals [9,10,11].

Near-infrared (NIR) analysis is a dynamically evolving, effective method for biological material characterization, owing to its rapidity, non-destructive nature, and low requirements with respect to sample preparation.

NIR spectroscopy is applied not only for quality control (QC) but also in process analytical technology (PAT) in various fields, such as in agriculture, in the food industry, in investigation of bioactive materials, in the pharmaceutical industry, and in the medical industry.

Molecule groups containing hydrogen (such as C–H, O–H, C=O, and N–H bonds) are infra-active; they will have a measurable NIR spectrum, resulting in a large range of organic materials suitable for NIR analysis in the NIR wavenumber regions (12,500–3800 cm^−1^) [12].

In the case of horticultural samples, it is particularly advantageous that NIR spectroscopy is also suitable for the direct analysis of solid samples. In the case of horticultural samples, it is particularly advantageous that NIR spectroscopy is also suitable for the direct analysis of solid samples. In most cases, only drying is used as sample preparation. This is necessary because the moisture content of horticultural samples is high and a very significant signal of water is given in the wavenumber range of 5200–5150 cm^−1^ (combination of O–H stretch and O–H deformation, O–H bend second overtone) and 6900–6800 cm^−1^ (O–H stretch first overtone) [13].

The methods of characterization of the species *Pleurotus ostreatus* are summarized in Table 1.

Due to growing consumer demand, there is an increasing emphasis on the cultivation of mushroom cultivars in which essential macro-components, polyphenols, and antioxidant components are present in appropriately high amounts [2,3,4].

The aim of our research was to offer mushroom growers a fast, inexpensive, non-destructive analytical method to select the cultivar with the most favorable properties for all free amino acids, TPC and FRAP values.

## 2. Materials and Methods

### 2.1. Materials

The processed and analyzed samples were from a mushroom breeding program. Different interspecific hybrids of *Pleurotus ostreatus* species were used for cultivation. The lack of available gene sources of the taxa leads to novel hybrids which are usually closely related. The mushroom mycelia were mixed into pasteurized wheat-straw based compost and cultivated indoors. After 16 days of spawn-running (25–26 °C compost temperature; 87–92% relative humidity (RH); 6000–8000 ppm CO_2_) the mushroom bags were cooled down to 17–18 °C with <1000 ppm of CO_2_ in air at 85–88% RH [31]. Depending on the mushroom strains used, 22–28 days later at spawning mushroom fruit bodies were harvested at ’medium’ stage [32] and processed immediately. All the mushroom samples were cultivated under same environmental parameters and harvested at same stage, so any of the differences in their chemical composition might be a result of their genetically distance.

The 14 varieties were cultivated at the same time and under the same conditions. The cultivation experiment was repeated at three consecutive time points. The harvested mushrooms were picked at the same phenological stage. A total of 250–300 g of fresh mushroom from the first flush was prepared for chemical measurements. Each sample was divided into three parts, so we had three parallel samples from each cultivation experiment. After repeating the cultivation experiments, we finally had nine samples from the same cultivars. Altogether, 126 samples were processed during the experiments. The averages of the parallel samples of each cultivars were taken into account in the statistical evaluation.

All reagents (high purity analytical reagents) were ordered from Reanal (Budapest, Hungary). Ultra-pure water (18.2 MΩ cm) was purchased from a Milli-Q system from Merck Millipore (Milford, MA, USA)

### 2.2. Methods

#### 2.2.1. Sample Preparation

The samples were dried under gentle conditions (60 °C; 19 h) in an air-conditioned oven (Memmert, Schwabach, Germany) and then ground to a particle size < 315 μm (Bosch TSM6A017C, Bosch GMbH, Stuttgart, Germany).

All our measurements were performed with the dried samples.

#### 2.2.2. Free Amino Acid Content (fAA)

The samples in three parallels (100 mg dried mushroom) were extracted with 5 mL cold 10% trichloroacetic acid (TCA) for 1 h on a shaker (Laboshake LS 500i, C. Gerhardt GmbH & Co. KG, Germany). Afterwards, the samples were filtered through a 0.2 μm pore membrane filter (Sartorius AG, Germany) and stored at −20 °C. The determination of different amino acids was accomplished with an automatic amino acid analyzer (Amino Acid Analyzer AAA400, Ingos Ltd., Praha, Czech Republic) furnished with an Ostion LCP5020 cation-exchange column (200 × 3.7 mm). They were separated by stepwise gradient elution using an Li–citric buffer system. At the end, the colorimetric detection was computed after post-column derivatization with ninhydrin reagent at 570 and 440 nm.

#### 2.2.3. Total Polyphenol Content (TPC)

The dried, ground samples were water-extracted, filtered, and stored at −26 °C. Total polyphenol content (TPC) was determined by the method of Singleton and Rossi with Folin–Ciocâlteu reagent [33].

The measurements were performed at pH = 10.0. The color change during the Mo(VI) yellow → Mo(V) blue redox reaction was detected spectrophotometrically (λ = 760 nm). At the applied wavelength, the light-absorbance of interfering components are negligible. The results were expressed as gallic acid equivalent (μmol GAE/g dry matter). The method is suitable to determine the total polyphenol content, but unsuitable to individual components determination.

#### 2.2.4. Antioxidant Capacity (Ferric Reducing Ability of Plasma—FRAP)

The dried, ground samples were water-extracted, filtered, and stored at −26 °C.

The total antioxidant capacity of the different extracts was measured by the FRAP method of Benzie and Strain [34]. According to the measurement principles, the ferric-2,4,6-tris(2-pyridyl)-S-triazine (TPTZ) complexes were reduced by reductive compounds in a pH = 3.6 environment. The reaction causes a blue color shift, detectable spectrophotometrically at λ = 593 nm. The results were expressed as ascorbic acid equivalent (µmol AAE/g dry matter).

#### 2.2.5. Fourier-Transform Near Infrared Spectroscopy (FT-NIR)

The dried, ground samples were subsequently measured with a rotatable quartz sample-container (d = 85 mm, 2 cm thick layer) on a Bruker MPA Multipurpose FT-NIR analyzer (Bruker Optik GmbH, Ettlingen, Germany). Spectra were collected in the 12,500–3800 cm^−1^ range using OPUS 7.2 (Bruker Optik GmbH, Ettlingen, Germany) software. Absorption spectra were recorded in diffuse reflectance mode. The resolution was 8 cm^−1^. The scanner speed was 10 kHz and each spectrum was the average spectrum of 32 subsequent scans. The internal background was measured using the integrated gold-coated surface of the integrating sphere. Five spectra were recorded for each sample. During the recordings, the dried, ground sample was mixed in the sample holder.

### 2.3. Chemometrics

#### 2.3.1. Kruskal–Wallis Test

The Kruskal–Wallis H test is a rank-based nonparametric test that can be used to determine whether there are statistically significant differences between three or more groups of an independent variable on a continuous or ordinal dependent variable.

The test does not identify where the differences occur or how many differences actually occur. Therefore, a test procedure for making pair-wise comparisons is needed. A common procedure used with the Kruskal–Wallis method is the Conover–Inman post-hoc procedure [35].

#### 2.3.2. Principal Component Analysis—PCA

This data reduction method is used primarily for the determination of spectral outliers in NIR studies. PCA is a linear unsupervised pattern recognition technique [36,37].

Second derivative data pre-processing was applied before principal component analysis (PCA).

#### 2.3.3. Linear Discriminant Analysis—LDA

LDA is a supervised classification algorithm. It can be used in order to identify which are the most significant features of a spectrum. LDA can also be used for classifying unknown samples [38].

For linear discriminant analysis, the original spectra were used, and there was no data pre-processing.

## 3. Results

### 3.1. Chemical Measurements

#### 3.1.1. Free Amino Acid Content (fAA)

Concentrations of the following amino acids were measured during the research: Alanine (Ala), Arginine (Arg), Asparagine (Asn), Aspartic acid (Asp), Cysteine (Cys), Glutamine (Gln), Glutamic acid (Glu), Glycine (Gly), Histidine (His), Isoleucine (Ile), Leucine (Leu), Lysine (Lys), Methionine (Met), Ornithine Orn), Phenylalanine (Phe), Proline (Pro), Serine (Ser), Threonine (Thr), Tyrosine (Tyr), Valine (Val).

In addition to the listed components, the concentrations of γ-aminobutyric acid (GABA), Cystathionine (Cyn) and 1-methylhistidine (HME) were also monitored.

The total free amino acid (fAA) concentration of cultivars is summarized in Figure 1 and Appendix A. The values of the individual amino acid components of the samples and the concentrations of GABA, Cyn and HME are summarized in Appendix A.

Based on the Shapiro–Wilk test, the measurement data were not normally distributed, therefore the significance test was performed with the Kruskal–Wallis test. Conover–Iman post-hoc test was applied to all possible pairwise comparisons between groups. Multiple pairwise comparisons using the Conover–Iman procedure (two-tailed test) were interpreted as well in Figure 1 (Appendix A). “A-H” notations refer to groups that can be formed based on Kruskal-Wallis statistical analysis. These classifications are the result of several pairwise comparisons established by the statistical program based on the sum of ranks and mean of ranks values.

Based on the results of the Kruskal–Wallis test of free amino acid content, it can be concluded that the cultivars Po_1–Po_2–Po_7 and Po_2–Po_4–Po_7 can be identified as different. Cultivar Po_2 has significantly the highest free amino acid concentration.

Using the Kruskal–Wallis test for each of the amino acid components tested, similar results were obtained in many cases for example, in the case of serine, threonine, histidine, 1-methyl histidine and arginine. Comparing the concentrations of leucine and tyrosine, the difference was significant for all candidates except Po_6.

Multiple pairwise comparisons using the Conover–Iman procedure (two-tailed test) of the limiting amino acids (cysteine (a), methionine (b)) are summarized in Figure 2 (Appendix A).

In some samples (Po_1, Po_3, Po_7, Po_9, Po_10, Po_12 and Po_14), the cysteine concentration was below the detection limit (1 μg/100 g). It can also be stated that the cultivars Po_2–Po_5–Po_6 and Po_8 can be identified as different. Regarding the concentration of methionine Po_2–Po_8 and Po_2–Po_10 cultivars can be identified as different.

#### 3.1.2. Total Polyphenol Content (TPC)

Comparing TPC values, several groups can be distinguished (Figure 3 and Appendix A). The following can be identified as significantly different candidates: Po_2–Po_3 (Po_4)–Po_5–Po_8–Po_10–Po_12 and Po_13 (Po_14). Po_5 cultivar had the highest TPC concentration.

#### 3.1.3. Antioxidant Capacity (Ferric Reducing Ability of Plasma—FRAP)

For FRAP values, in contrast to TPC data, fewer significantly different samples could be distinguished (Figure 4 and Appendix A). These were as follows: Po_2–Po_10; Po_2–Po_14; Po_5–Po_10 and Po_5–Po_14. Of the cultivars, Po_2 and Po_8 had the highest FRAP values.

### 3.2. FT-NIR Measurements

#### 3.2.1. Spectrum Profiling

Comparing the NIR spectra, minimal differences can be observed in the traditional spectrum image. Baseline shift is a consequence of scatter due to different particle sizes. The subtle, characteristic differences become visible primarily in the second derivative curve (Figure 5).

For better clarity, some details of the measurement range were magnified, and the spectra were separated (Figure 6 and Figure 7). Within the 9000–5400 cm^−1^ interval, an important characteristic difference between the curves is observed in the areas 7650–7090 and 6890–6630 cm^−1^. This discrepancy can clearly be explained by a qualitative difference.

The most striking difference is observed in the area of 7650–7090 cm^−1^. Based on the characteristics of the curves, two typical groups can be distinguished. The first group includes the spectra of Po_1, Po_2, Po_4, Po_5, Po_6, Po_7, Po_9, Po_11, Po_13 and Po_14, while the second group includes the spectra of Po_3, Po_8, Po_10 and Po_12 (Figure 6). The signal of aliphatic and aromatic hydrocarbon chains can be detected primarily in this area (Table 2).

In the case of the first group, the peak at 7100 cm^−1^ is typical (dashed line), while in the second group, the peak at 7230 cm^−1^ is observed (dashed-dotted line).

A smaller difference is observed in the wavenumber interval 6890–6630 cm^−1^. In this area, there is mainly a difference in peak height, but this cannot be explained by a qualitative difference. These spectral differences are clearly observed in the characteristic areas of hydrocarbon chains, phenolic structures, and alcoholic OH (Table 2). Given that neither the TPC values (Figure 3, Appendix A) nor the FRAP values (Figure 4, Appendix A) of these four cultivars are outstanding, it can be reasonably assumed that their polyphenolic compounds are qualitatively different from the other ten cultivars.

In the wavelength range 5400–3800 cm^−1^ (Figure 7), further differences between the spectra can be observed. The initial peak is seen at 5314 cm^−1^ for samples Po_3, Po_8, Po_10 and Po_12 (dashed-dotted line), while for the other cultivars this peak shifts to 5290 cm^−1^ (dashed line). Given that the samples were dried and not lyophilized, although the drying conditions were the same, it can be assumed that this initial peak shift may be due to the formation of water molecules remaining after drying and the formation of any hydrogen bonds.

Further differences are observed for cultivars Po_2, Po_3, Po_6, and Po_10 in the region 4400–4100 cm^−1^ (dotted line, Figure 7) (Table 3).

The wavenumber range 4400 to 4100 cm^−1^ is the absorption area of many characteristic functional groups. In addition to proteins, the characteristic peaks of fats and carbohydrates (including fibers) also appear here. In our previous studies, we have already established that there is no significant difference between the cultivars in terms of carbohydrate content [40].

The differences observed here (Figure 7) can be explained clearly by the different ratio of free amino acid components. The Po_2 sample is characterized by the highest total free amino acid concentration (Figure 1). The most significant difference is observed for sample Po_2. In our opinion, such a characteristic difference in the spectrum can be explained by the fact that the individual concentrations of several amino acid components (Asp, Thr, Ser, Pro, Gly, Ala, Val, Met, Ile, Leu, Tyr, Phe, His, Arg, HME) are the highest in this sample.

The spectrum of the Po_3 sample has similar characteristics to that of the Po_2 sample. This can be explained by the fact that for several amino acid components (Ser, Thr, Ala, His, Arg, HME) there was no significant difference between the two cultivars.

When analyzing the spectra, it must be emphasized that in the NIR range, we cannot perform qualitative identification in the strict sense on the basis of the spectral images. Based on the characteristic absorption bands, we can infer the presence of proteins and carbohydrates, but these bands broaden and overlap, so we cannot connect a peak to a specific compound. In our research, we basically looked for the presence of proteins (free amino acids), phenolic/polyphenolic components, and carbohydrates responsible for antioxidant effects.

#### 3.2.2. Chemometric Evaluation of NIR spectra

Principal Component Analysis (PCA)

PCA analysis of the spectra as an unsupervised pattern recognition method examined. PCA analysis of the spectra as an unsupervised pattern recognition method examined whether any pattern between the spectra could be recognized. Given that during the comparison of the second derivative curves in the spectral studies (Figure 6 and Figure 7), we observed that four cultivars—Po_3, Po_8, Po_10 and Po_12—differed from the others, before PCA a second derivative as a data preprocessing technique was applied (Figure 8a,b). PCA analysis was performed with UNSCRAMBLER 10.4 (CAMO, Oslo, Norway) software.

As a result of PCA, the same four cultivars could be distinguished, which were indicated by examination of the spectra.

The results of the cluster analysis supported the visually perceived discrepancy and the PCA result (Figure 9).

Linear Discriminant Analysis (LDA)

The applied LDA supervised pattern recognition is based solely on the analysis of NIR spectral data with STATISTICA™ (Tulsa, OK, USA).

Based on our reference measurement results, we formed groups based on total free amino acids, cysteine, methionine proline (Figure 10), as well as TPC and FRAP values (Figure 11).

These amino acid components were chosen because cysteine and methionine are limiting amino acids, whereas proline concentration is correlated with stress effects.

Forward stepwise model building method and threefold cross-validation were applied in the evaluation. In the first step, the three parallel spectra of the samples from 12,500 to 4000 cm^−1^ were used for principal component analysis. We did not apply data preprocessing. The first ten PCA scores were used for the further analysis with LDA. Proper validation should test whether the results are artefacts or not. For this purpose, as another validation method for the model, X-scrambling randomization test was used three times [41].

The classification was 100% successful for free amino acids. Due to some misclassifications, we achieved a probability of nearly 93% for cysteine, nearly 90% for methionine and nearly 95% for proline.

When examining the TCP and FRAP values, the pattern was recognized with a probability of more than 83% for both parameters (Figure 11).

## 4. Discussion

Based on our experimental results, the Po_2 cultivar clearly stood out from the 14 *Pleurotus* cultivars in terms of free amino acid content. This cultivar had a significantly higher concentration not only of the free amino acid but also for its individual amino acid components.

Based on total polyphenol values, cultivars Po_5, Po_6, and Po_7 had the highest concentration, while for FRAP values, cultivars Po_2 and Po_5 showed the highest.

Based on the analysis of the FT-NIR spectra, the cultivars Po_3, Po_8, Po_10, Po_12 can be clearly distinguished in the wavelength range of 7420–7067 cm^−1^ compared to the other cultivars. The spectral difference is explained by the presence of different qualities or types of polyphenols and reducing (antioxidant) components.

The characteristic separation was also supported by our PCA and cluster analyzing result.

Successful classification was performed using LDA to analyze total free amino acids, cysteine, methionine (limiting amino acids), and proline content (stress-related amino acid).

Based on our results, it can be stated that without chemical analysis, we performed a successful LDA assay with 100% accuracy for all free amino acid concentrations. Examining the cysteine content, the classification was successful with an accuracy of 92.86%, for methionine 89.74% and for proline 94.87%.

Regarding the TPC and FRAP values characteristic of stress resistance, the grouping was achieved with an accuracy of 88.1% and 83.3%, respectively. Although 90% accuracy was not achieved for any of the parameters in this case, we still consider our results to be excellent, considering that both the polyphenolic compounds and the compounds responsible for the reducing effect can be very diverse.

Based on our results, it can be stated that PCA and LDA methods based solely on the analysis of FT-NIR spectral data can be successfully used to classify *Pleurotus ostreatus* cultivars for total free amino acid, cysteine, methionine, and proline concentrations, as well as TPC and FRAP values.

Based on the results, we have to state that we cannot predict the values of the examined parameters, but based on the spectral data we can classify the samples into a group that can be characterized by a chemical result.

Thus, FT-NIR spectroscopy offers the possibility of non-destructive, fast qualification.

## Figures and Tables

**Figure 1 sensors-20-06632-f001:**
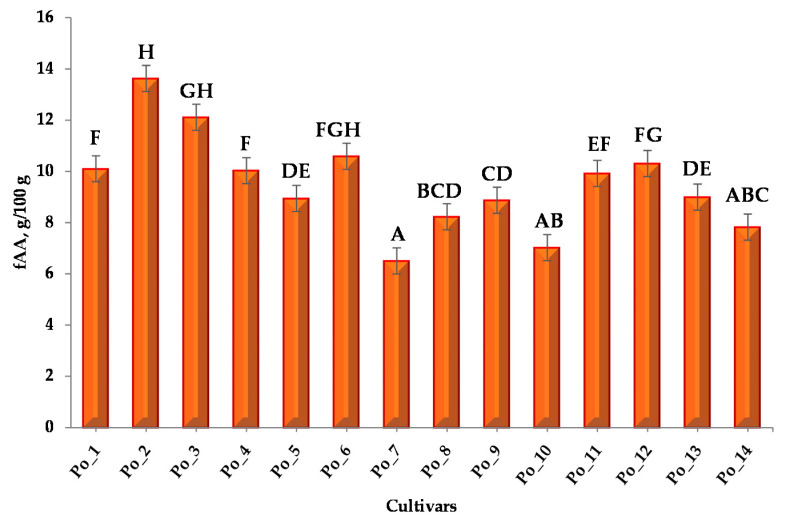
Total free amino acid (fAA) concentration and multiple pairwise comparisons of cultivars.

**Figure 2 sensors-20-06632-f002:**
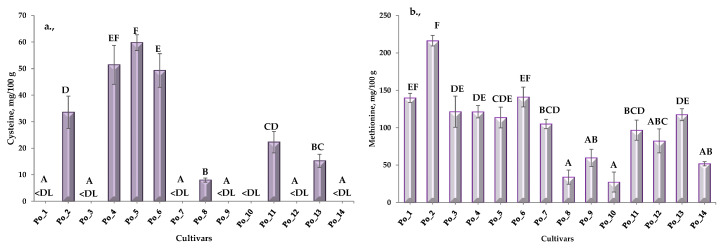
Cysteine (**a**) and methionine (**b**) concentration and multiple pairwise comparisons of cultivars (DL = 1 μg/100 g). (“A–F” = Conover-Iman post hoc test groups).

**Figure 3 sensors-20-06632-f003:**
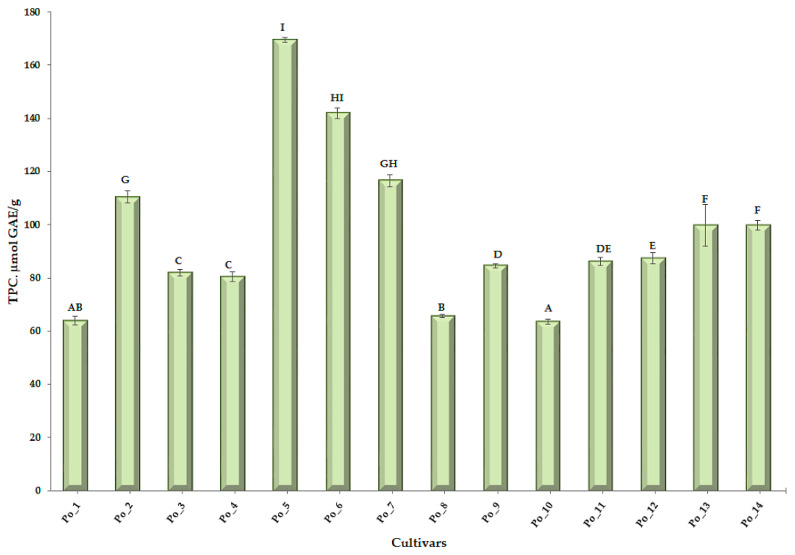
Total polyphenol content (TPC) and multiple pairwise comparisons of cultivars. (“A-H” = Conover-Iman post hoc test groups).

**Figure 4 sensors-20-06632-f004:**
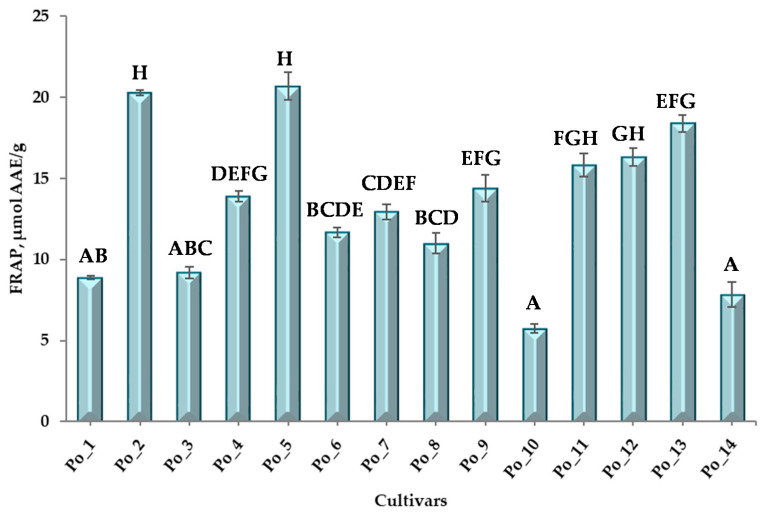
FRAP values and multiple pairwise comparisons for FRAP values of cultivars. (“A-H” = Conover-Iman post hoc test groups).

**Figure 5 sensors-20-06632-f005:**
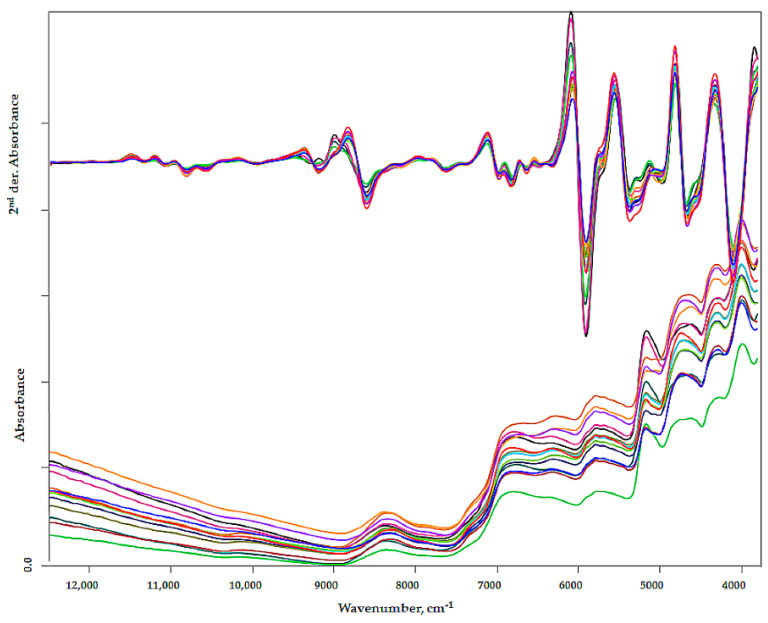
FT-NIR spectra of differential cultivars and their second derivative.

**Figure 6 sensors-20-06632-f006:**
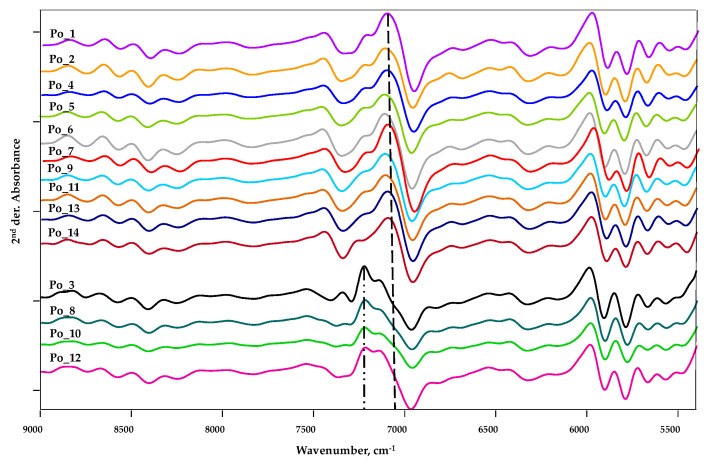
FT-NIR spectra in measurement range 9000–5400 cm^−1.^

**Figure 7 sensors-20-06632-f007:**
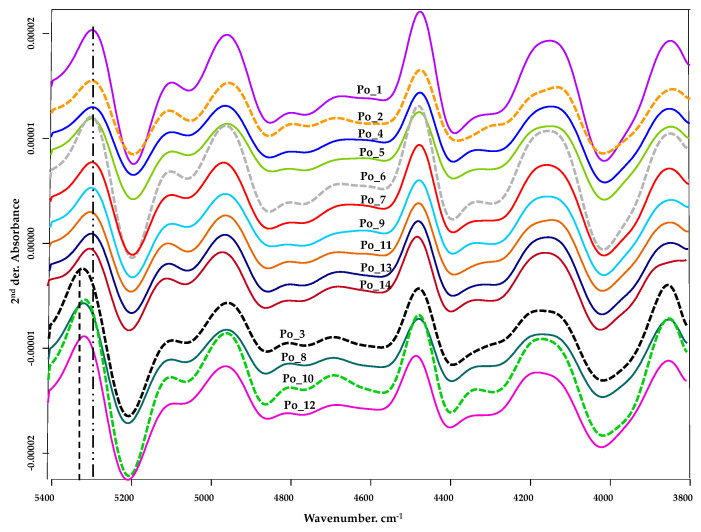
FT-NIR spectra in measurement range 5400–3800 cm^−1.^

**Figure 8 sensors-20-06632-f008:**
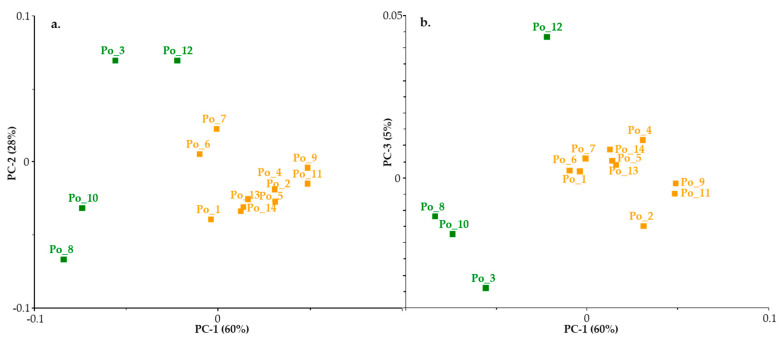
Principal component analysis (PCA) test result PC1 vs. PC2 (**a**) and PC1 vs. PC3 (**b**).

**Figure 9 sensors-20-06632-f009:**
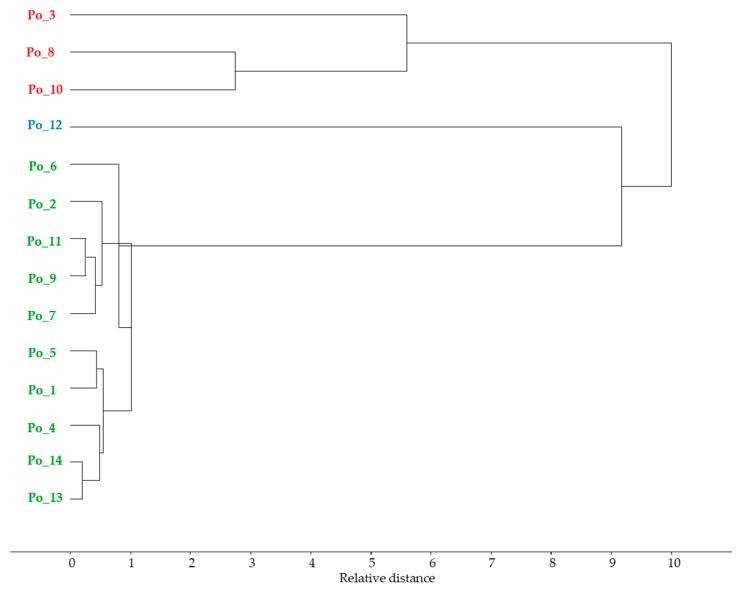
Median linkage clustering using Squared Euclidean distance.

**Figure 10 sensors-20-06632-f010:**
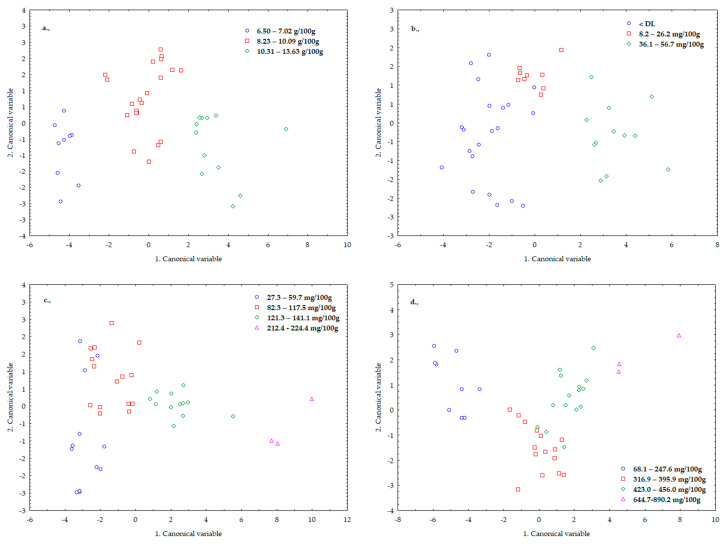
Linear discriminant analysis (LDA) results for total free amino acids (**a**), cysteine (**b**), methionine (**c**) and proline (**d**) content.

**Figure 11 sensors-20-06632-f011:**
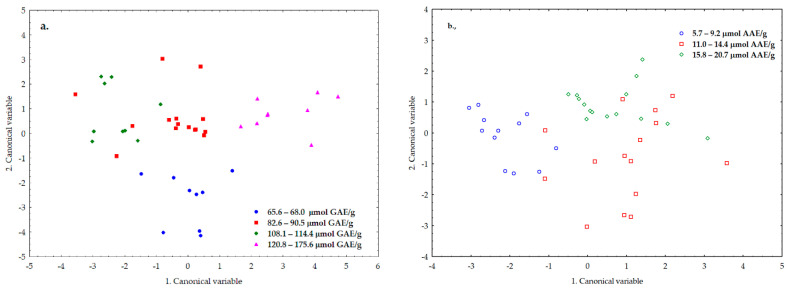
LDA results for TPC (**a**) and FRAP (**b**) value.

**Table 1 sensors-20-06632-t001:** Summary of the examples for the determination of amino acid content, antioxidant capacity (FRAP), total phenol content (TPC) and classification of *Pleurotus ostreatus.*

Examined property	Method	References
Free radical scavenging activity	DPPH value	[2,7,14,15,16,17,18,19],
	FRAP value	[2,19,20]
Total polyphenol content	Folin Ciocalteu method	[2,17,18,19]
Active antioxidant polysaccharides	HPLC	[15]
Amino acid content	HPLC with fluorescence detector	[2,21,22,23,24,25,26]
Glucans and ergosterol	ATR-FTIR	[7,8]
Strain classification and taxa delimitation	FT-IR	[27]
Identifying bioactive compounds	ATR-FTIR	[27]
Antioxidant activity	DPPH value and FTIR	[28]
Water soluble polysaccharides	FTIR	[29]
Phylogenetic relatedness—DNA	FTIR	[30]

DPPH: 2,2-diphenyl-1-picryl-hydrazyl; FRAP: Ferric Reducing Ability of Plasma; ATR-FTIR: Attenuated total reflection Fourier-transform infrared spectroscopy; FT-IR: Fourier-transform infrared spectroscopy.

**Table 2 sensors-20-06632-t002:** Spectra structure correlations in 7353–6900 cm^−1^ region [13,39].

Wavenumber, cm^−1^	Spectra Structure	Material Type
7353	C–H methyl second combination band	Aliphatic hydrocarbons
7300	Combination of first overtone of methyl C–H stretching and CH_3_ bending	Aromatic hydrocarbons
7194	Combination of first overtone of methyl C–H stretching and CH_3_ bending	Aliphatic hydrocarbons
7168	C–H methylene second combination band	Aliphatic hydrocarbons
7163	Combination of first overtone of methyl C–H stretching and CH_3_ bending	Aliphatic hydrocarbons
7200–7100	First overtone of O–H stretching modes	Alcohols
7140–6940	First overtone of O–H stretching modes from phenols	Phenolic O–H
7085–7067	Combination of first overtone of methylene C–H stretching and methylene C–H bending modes	Aliphatic hydrocarbons
7057	Combination of first overtone of C–H and aryl C–H	Aromatic hydrocarbons

**Table 3 sensors-20-06632-t003:** Spectra structure correlations in 4400–4100 cm^−1^ region [13,39].

Wavenumber, cm^−1^	Spectra Structure	Material Type
4400	CONH_2_ specifically due to peptide β-sheet structures	Proteins
4365–4370	CONH_2_ specifically due to the α-helix peptide structure	Proteins
4360	First overtone of CH stretching and C–H bending modes	C–H aryl
4348	Second overtone of C–H stretching mode	Amides
4333	Combination of first overtone of CH_2_ asymmetric stretching and CH_2_ bending modes	Aliphatic hydrocarbons
4314	Combination of first overtone of CH_2_ asymmetric stretching and CH_2_ bending modes	Aliphatic hydrocarbons
4261	Combination of first overtone of CH_2_ symmetric stretching and CH_2_ bending modes	Aliphatic hydrocarbons
4232	Combination of first overtone of CH_2_ symmetric stretching and CH_2_ bending modes	Aliphatic hydrocarbons
4190	C–C bending and C–H stretching modes	C–H Aryl
4155	C–H stretching and C–H bending modes	C–H Aryl
4091	C–C stretching and C–H stretching modes	C–H Aryl
4090	CONH_2_ specifically due to the α-helix peptide structure	Proteins

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
