# Peer review of "Application of Near-Infrared Spectroscopy to Investigate Some Endogenic Properties of Pleurotus ostreatus Cultivars"

_sensors, 2020, doi:10.3390/s20226632_

Round 1

Reviewer 1 Report

This manuscript by Fodor et al. tested 14 different Pleurotus ostreatus for free amino acid content, total polyphenol content, and antioxidant capacity in order to select the cultivars with the most favorable traits.

This is a very interesting study, in particular because it aims to elucidate chemical information about the amino acids present in the samples. This goal distinguish this work and research performed at this direction is very much needed, as NIR spectroscopy is still struggling in the chemical specificity/selectivity. This work is based on an adequate experimental material and the data-analytical results seem to be sufficient. The selection of methods and the execution of the study are correct. The manuscript is well organized and nicely written. There are only minor shortcomings that could be improved within a minor revision.

  1. How many spectra per sample were measured? Just a single spectrum? More details on spectral measurements are needed (resolution, scan numbers, measurement repetitions, e.g. triplicate)
  2. The Introduction is very short and focused on the information on the samples. Perhaps few sentences about NIR spectroscopy as the technique used for the study would be beneficial.
  3. There is no specification of the reagents used in the study (only ‘high purity analytical reagents’).
  4. Only in Abstract the number of samples is mentioned (14?). In principle, all the information given in the Abstract should be also present in the main body of the article.
  5. PCA should be resolved as ‘principal component analysis’ not ‘principal components analysis’
  6. “In this area, there is mainly a difference in peak height, but this cannot be explained by a qualitative difference.” It remains unclear what is the intended meaning of “qualitative difference” in this context.
  7. It is not entirely clear what is the meaning of ‘’ABC’’, ‘’H’’ or ‘’F’’ on the figures?
  8. 7420 cm-1 is definitely (and by far) too low lying for this absorption to originate from the second overtone of O-H stretching modes from phenols, even if a red-shift resulting from hydrogen-bonding would be considered. This assignment is in sharp contradiction with the one provided in the same table for the ‘First overtone of O-H stretching modes from phenols’ (7140 – 6940 cm-1). In general, the assignments could be refined. Recent literature shows great increase in the NIR band assignments (DOI: 10.3389/fchem.2019.00048).
  9. The information in Table 3 and the discussion in the text may make an impression that the “characteristic bands” may be used for identifying the chemical constituents present in the studied sample, e.g. proteins (akin to IR spectroscopy). However, the chemical specificity and selectivity of NIR spectra is obviously much lower as the result of the extensive band overlapping. Therefore, while of course the bands due to e.g. proteins contribute to the absorption bands observed in NIR spectra in a given region, it is necessary to remember that it is most often impossible to elucidate individual contributions to the observed NIR spectral lineshape. This phenomenon is very well demonstrated in the literature mentioned in the point above (DOI: 10.3389/fchem.2019.00048).
  10. The second part of the sentence where the description of PCA is given (“pattern between the spectra could be recognized.”) could be improved by better reflecting the key points of the actual algorithm (e.g. minimization of variance).
  11. The Discussion (Section 4) is interesting albeit perhaps a bit too short. However, as the manuscript is a letter, it may be acceptable.
  12. English:
  • For principal components analysis (PCA) was second derivative pre-processing of spectra applied.
  • Page 1, Line 29- remove additional ‘.’
  • In the caption to Table 3 “interval” would sound more natural if replaced by “region”

Author Response

Dear Reviewer,

In the name of my colleagues I would also like to thank the Reviewer for her/his valuable comments and recommendations. All comments and suggestions have been accepted and the manuscript has been modified accordingly. All modifications are marked in red, and all comments have been responded point by point.

Reviewer 2 Report

The work presented in this manuscript offers mushroom growers a fast, inexpensive, non-destructive analytical method to select the cultivar with the most favourable properties for all free amino acids,TPC and FRAP values.

The work manuscript is well written and presented. The outcome of this research with more validation can be easily implemented into the ongoing quality control protocols for mushroom producers and retailers in terms of defining their products as if highly nutritional value.

Please find a word version of the manuscript with my feedback via track changes.

Once this feedback is addressed appropriately, I feel this manuscript can be accepted for publication.

Author Response

Dear Reviewer,

In the name of my colleagues, I would also like to thank the Reviewer for her/his valuable comments and recommendations. All comments and suggestions have been accepted and the manuscript has been modified accordingly. All modifications are marked in red, and all comments have been responded point by point.

Round 2

Reviewer 1 Report

The revision of the manuscript by Fodor et al. was performed well. I recommend to accept this manuscript in present form.

Reviewer 2 Report

Authors have duly responded to all the feedback provided.